# 3D HUMAN RECONSTRUCTION IN THE WILD WITH SYNTHETIC DATA USING GENERATIVE MODELS

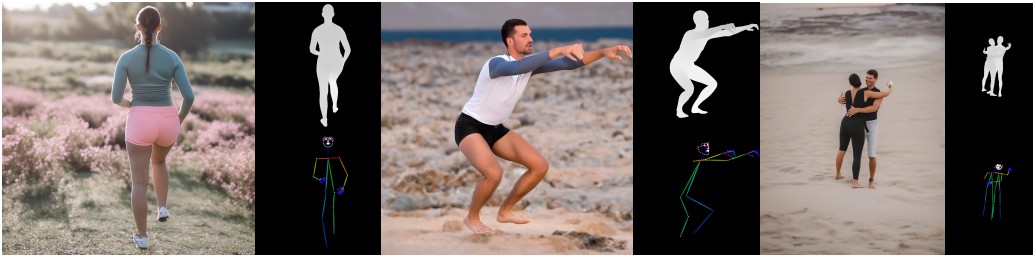

Figure 1: `Pose++` generates diverse photo-realistic human images and corresponding body annotations, *e.g.* 2D landmarks and 3D meshes, with a multi-condition diffusion model.

## ABSTRACT

Human pose and shape estimation from monocular images play a fundamental role in computer vision applications such as augmented reality, virtual try-on, and human motion analysis. However, large-scale human datasets in the wild with 3D ground-truth annotations are very difficult to obtain. Previous high-quality 3D human pose datasets are usually obtained by either motion capture devices or computer graphics rendering techniques, both are expensive and laborious. In this work, we propose an effective approach based on recent diffusion models, termed `Pose++`, which can effortlessly generate human images and corresponding 2D human skeletons and 3D mesh annotations. Specifically, we first leverage a multi-conditioned stable diffusion model to generate diverse human images and initial ground-truth labels. At the core of this step is that we can easily obtain numerous depth and keypoints conditions from a 3D human parametric model, *e.g.*, SMPL-X, by rendering the 3D mesh onto the image plane. The generated human image and the corresponding 3D mesh with camera parameters can be regarded as a pair of training samples. As there exists inevitable noise in the initial labels, we then cast the problem into a label-denoising process by exploiting an off-the-shelf 2D human pose estimator to filter negative data pairs and further optimize the pose parameters. Finally, we can build a unified human pose dataset with both 2D skeleton and 3D parametric model annotations. Experiments on 2D datasets (COCO, OCHuman) and 3D datasets (3DPW, RICH, SSP-3D) demonstrate the effectiveness of our approach. Thus, our method offers a promising avenue for advancing the field of human pose and shape estimation by generating large-scale human images and high-quality annotations in a fully automated fashion.

## 1 INTRODUCTION

Estimating human pose and shape (HPS) (Kanazawa et al., 2018; Lin et al., 2021; Li et al., 2021b; 2022) from a single RGB image is a core challenge in computer vision and has many applications in robotics, computer graphics, and digital content creation. Current HPS estimation methods require well-annotated datasets to achieve good performance. Unfortunately, collecting large-scale human body data is time-consuming and expensive.

As shown in Table 1, There are mainly two types of pipelines for capturing accurate 3D human body data. The first type is the indoor mocap systems, *e.g.* marker-based systems, and vision-

| Data Type | | Settings | | | | |
|---|---|---|---|---|---|---|
| | | Assets | Human Workload | Comp. Cost | Scale-Up Diff. | Magnitude |
| Real-World | MoCap | mocap system | Actors | $\times$ | hard | $1 \times 10^5$ |
| | MV. Pseudo | RGB(D) cameras | Annot./Cam. Calib. | Models/Optim | hard | $1 \times 10^4$ |
| | Mono. Pseudo | RGB(D) cameras | Annot./Cam. Calib. | Models/Optim | medium | $1 \times 10^5$ |
| Synthetic | | 3D Avatars/Scenes | Technical Artists | Render | easy | $1 \times 10^6$ |
| Generated | | $\times$ | $\times$ | Models/Optim | none | $\infty$ |

Table 1: 'MV.' and 'Mono.' stands for 'multi view' and 'Monocular' separately. 'Annot.' and 'Cam. Calib.' stand for 'anntation' and 'camera calibration' separately. 'Comp. Cost' and 'Scale-up Diff.' stands for 'computation cost' and ''scale-up difficulty' separately.

based systems. Many existing datasets (Ionescu et al., 2013; Tripathi et al., 2023; Cai et al., 2022; Mehta et al., 2017) use this pipeline to capture human body attributions. However, the pipeline suffers from four drawbacks: 1). The mocap systems are expensive. 2). the synchronization and operation of the system are complicated. 3). The number of actors in the dataset is limited. 4). The background is typically the indoor or laboratory environment, making large-scale human data with versatile scenes infeasible. The second type is synthesizing 3D human datasets using computer graphics (CG) rendering (Black et al., 2023; Wood et al., 2021; 2022). The drawbacks of this pipeline are three-folds: 1). High-quality 3D assets, including drivable avatars and scene assets, are expensive. Wood et al. (2021; 2022) do not open-source their data generation pipeline for sake of the commercial purpose. 2). Special knowledge of 3D rendering is required, making it impossible to use cheap crowdsourcing platforms like Amazon Mechanical Turk. 3). The domain gap between the rendered images and real-world images is non-negligible. As mentioned in Black et al. (2023), the HPS accuracy trained on rendered images depends on backbone pre-training, especially 2D COCO keypoint dataset pre-training. This pheromone suggests that the synthetic data still has room for improvement in terms of realism.

As it is hard to obtain large-scale 3D human pose datasets in the wild, some researchers have considered leveraging existing large-scale 2D human pose datasets by optimization-based and weakly-supervised methods. SMPLify (Bogo et al., 2016) proposed to fit the parameters of a 3D human model to the location of 2D keypoints. EFT (Joo et al., 2021) introduced the Exemplar Fine-Tuning strategy by overfitting a pre-trained 3D pose regressor with 2D keypoint reprojection loss, taking the final output of the regressor as pseudo labels. However, these methods still suffer from poor performance on 3D human pose benchmarks.

In this paper, we address these limitations by proposing a new data generation pipeline, termed `Pose++`, that can simultaneously generate photo-realistic human images in the wild, as well as corresponding well-aligned 2D human skeletons and 3D mesh annotations in a fully automatic fashion. The challenge of the pipeline lies in two folds. On one hand, *how to ensure the pose, shape, and scene diversity of generated human images are critical in simulating real-world human distribution.* A naive solution is taking advantage of the text-to-image diffusion models, *e.g.*, Stable Diffusion (Rombach et al., 2022), by feeding different text prompts to the model and employing pre-trained pose estimators to get the pseudo labels. However, the text prompt alone is not fine-grained enough to create versatile human bodies. To solve this problem, we sample SMPL-X parameters of human bodies from large-scale human motion capture datasets (Mahmood et al., 2019; Black et al., 2023). Then, we render the human mesh into the depth map and keypoint heatmap with a random camera as the extra input conditions. Finally, we feed text prompts, depth map, and keypoint heatmap to a multi-conditioned ControlNet (Zhang & Agrawala, 2023) for generating human images. As such, we can get fine-grained control of the human body, and get initial training data pairs from the input conditions and output human images. On the other hand, *how to ensure the alignment between the human images and generated annotations is critical for the training of downstream tasks*. Experiments show that there exist label noises in the initial training data pairs. For example, the generated human and the input conditions form a mirror pair, or the human head orientation in the image is not consistent with the input SMPL-X parameters. To resolve this problem, we propose a two-stage label-denoising and refinement strategy. First, we use an off-the-shelf 2D pose estimator to filter the wrong-generated images by computing the average precision (AP) of symmetric joints. Second, we finetune the 2D keypoints in SMPL-X format to target 2D pose datasets with a transformer-based keypoint decoder. Upon getting the 2D keypoints, we optimize the head poses

of SMPL-X with EFT (Joo et al., 2021). With the aforementioned pipeline, we can get well-aligned training pairs and finally generate a large-scale 3D human dataset in the wild with around 600,000 samples, $1024 \times 1024$ resolution. Compared to previous datasets, our pipeline can generate diverse human identities and various in-the-wild scenes. Notably, the pipeline is much cheaper than both mocap-based and CG-based counterparts and is able to scale up 3D human datasets in the wild.

Our contributions can be summarised as follows. 1). We propose a fully automatic pipeline to synthesize realistic and diverse human images with well-aligned annotations, including 2D keypoints, 3D SMPL-X parameters, and text descriptions. The dataset can empower a wide range of downstream perception tasks by rendering SMPL-X mesh into corresponding annotation format, *e.g.*, human pose and shape estimation, human part segmentation, and human normal prediction. 2). We verify the quality of the generated dataset on 2D human pose estimation (HPE) and 3D human mesh reconstruction tasks. Experiment results show that the proposed pipeline can achieve comparable performance on several 2D HPE and 3D HPS benchmarks under the same settings.

## 2 RELATED WORKS

### 2.1 HUMAN POSE AND SHAPE ESTIMATION DATASETS

**Real-World Human Pose Data** is vital for accurate, realistic modeling in 3D Human Pose and Shape Estimation tasks. High-quality data is typically captured using advanced motion capture devices like Inertial Measurement Units (IMUs) (von Marcard et al., 2018; Mahmood et al., 2019; Sigal et al., 2010) or Optical sensors (Ionescu et al., 2013), designed to capture precise marker movements or joint rotations. However, their deployment can be burdensome due to factors such as cost, setup complexity, and space requirements. Responding to these challenges, research has explored alternative methods to capture pseudo labels from diverse image types, including RGBD (Hassan et al., 2019), multi-view (Cai et al., 2022), and single-view images (Bogo et al., 2016), eliminating the need for motion capture gear. SLOPER4D (Dai et al., 2023) consolidates data from IMU sensors, LiDAR, and RGB information to construct a large-scale urban human-scene dataset. Such methods often leverage perception models to derive 2D cues from images, which are further optimized by a 3D joint re-projecting loss.

**Synthetic Human Pose Datasets,** developed with computer graphic techniques, has been used for many years. SURREAL (Varol et al., 2017) applies human skin and cloth textures to bare SMPL meshes, which lack realistic details. AGORA (Patel et al., 2021) uses high-quality static human scans for image rendering, but this routine also suffers from a high workload of scanning and rigging. However, rendering realistic manipulatable synthetic human datasets involves many challenges, including the need for diverse virtual properties for realistic data. BEDLAM (Black et al., 2023) and Synbody (Yang et al., 2023) add varied hair models and skin textures to SMPL-X (Pavlakos et al., 2019) meshes and simulates physically plausible cloth and hair movements. These processes can be resource-intensive. Furthermore, the use of rendering engines demands many professional skills. Thus, the rendering process can be computationally expensive and time-consuming.

**Controllable Human Image Generation** has gained great traction with the advancement of Stable Diffusion (Rombach et al., 2022; Zhang & Agrawala, 2023). Text2Human (Jiang et al., 2022) uses a diffusion-based transformer sampler in response to text prompts and predicts indices from a hierarchical texture-aware codebook to conditionally generate realistic human images. HumanSD (Ju et al., 2023) introduces a skeleton-guided diffusion model with a novel heatmap loss for pose-conditioned human image generation.

**Generative Models for Perception Tasks.** Several studies have effectively utilized datasets generated by diffusion models for perception tasks. For instance, Voetman et al. (2023) employed these datasets to train detection models, and Azizi et al. (2023) demonstrated that classification models can achieve state-of-the-art results on ImageNet (Deng et al., 2009) when fine-tuned on generated images. StableRep (Tian et al., 2023) found that training modern self-supervised methods on synthetic images from Stable Diffusion Models can yield impressive results. The learned representations often surpass those learned from real images of the same sample size. DatasetDM (Wu et al., 2023) trained decoders using limited data and succeeded in decoding the rich latent code of the diffusion model as precise perception annotation. This has enabled the generation of an infinitely large annotated dataset, proven effective in segmentation, depth estimation, and 2D human pose estimation.

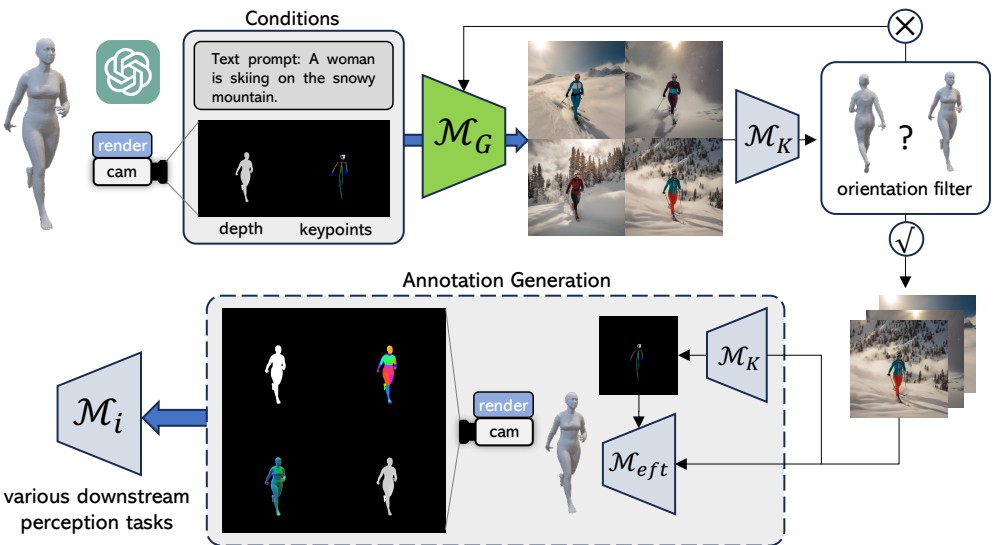

Figure 2: Full pipeline of automatic data generation. $\mathcal{M}_G$ indicates the ControlNet (Zhang & Agrawala, 2023). $\mathcal{M}_K$ denotes a pre-trained 2D pose regressor. $\mathcal{M}_{eft}$(Joo et al., 2021) denotes the 3D human pose regressor for label refinement.

DiffusionHPC (Weng et al., 2023), closely related to this paper, is pioneering in employing diffusion models to render human images for 3D HPS tasks. It leverages a pre-trained 3D pose regressor to estimate the human mesh, subsequently renders a depth map, and then leverages a depth-to-image diffusion model to generate human images. Different from DiffusionHPC, the input condition of Pose++ is sampled from large-scale motion datasets and we use multi-resource conditions to enhance the alignment between the generated images and 3D pose labels. Besides, we involve an extra refinement step to refine the initial 3D pose parameters.

# 3 METHOD

We present Pose++, a simple yet effective pipeline for creating versatile human body images and corresponding perception annotations in a fully automated fashion, which can be used for many downstream human perception tasks, such as 2D/3D human pose and shape estimation, human part segmentation, and human action recognition (see Fig. 2). The core idea of the proposed pipeline is creating large-scale image-mesh-caption pairs by incorporating off-the-shelf 2D generative models, *e.g.* Stable Diffusion (Rombach et al., 2022) and 3D human parametric models (Pavlakos et al., 2019). For the sake of completeness, we give a brief review of the controllable text-to-image (T2I), image-to-image (I2I) generative models and the 3D human parametric model, SMPL-X (Pavlakos et al., 2019) in Section 3.1. In the following subsections, we first illustrate how we generate the initial human image-annotation pairs in Section 3.2. Then we show how we refine the initial 2D keypoints labels and 3D pose labels to get high-quality training pairs in Section 3.3.

## 3.1 PREREQUISITES

**Stable Diffusion Models** (Rombach et al., 2022; Podell et al., 2023) are text-to-image diffusion models capable of generating near photo-realistic images given any text input. They have been revealed to be capable of synthesizing more diverse and higher-quality images compared to previous dominant GAN-based models (Goodfellow et al., 2016; Brock et al., 2018; Karras et al., 2019). Controllable image-to-image adapters are frameworks designed for empowering text-to-image diffusion models with more image-level control signals. ControlNet (Zhang & Agrawala, 2023) and T2I-Adapter (Mou et al., 2023) are two representative lightweight adapters that only apply several additional blocks to the original stable diffusion models. During the training of adapters, the text-to-image diffusion model is frozen. Thus, they significantly reduce the training cost while keeping the generation ability of the origin text-to-image models to the maximum extent.

**SMPL-X** (Pavlakos et al., 2019), defined as $M(\boldsymbol{\beta}, \boldsymbol{\theta}, \boldsymbol{\psi}) : \mathbb{R}^{|\theta| \times |\beta| \times |\psi|} \rightarrow \mathbb{R}^{3N}$, is a 3D wholebody human parametric model, employing shape, expression, and pose parameters to control the entire body mesh. The shape parameters $\boldsymbol{\beta} \in \mathbb{R}^{200}$ are dictated by the first 200 principal components of a linear shape space learned from scanned human meshes. The expression parameters $\boldsymbol{\psi} \in \mathbb{R}^{50}$ represent coefficients of a low-dimensional linear space, while the pose parameters model relative 3D rotations for 55 joints, encompassing the body, jaw, and hand poses. The function of SMPL-X provides a differentiable skinning process that uses pose, shape, and expression parameters as inputs and delivers a triangulated mesh $V \in \mathbb{R}^{N \times 3}$ with $N = 10475$ vertices. The reconstructed 3D joints $J \in \mathbb{R}^{144 \times 3}$ can be obtained using a forward kinematics process.

## 3.2 Initial Human Image and Annotation Generation

**Camera simulation.** One drawback of vision-based motion capture systems is that they need to calibrate and synchronize the camera's intrinsic and extrinsic parameters during the capturing. Thus, the generated human data are limited in terms of the scales and view diversity. On the contrary, our pipeline gets rid of the physical RGBD cameras and can simulate arbitrary human scales and body orientation. Specifically, we randomly determine the orthographic scale $s$ of the human body ($s \in [0.45, 1.1]$), along with the horizontal shift $(t_x, t_y)$ within a range of $[-0.4/s, 0.4/s]$. This methodology ensures that the majority of body parts are visible in the image. Following Kanazawa et al. (2018); Wang et al. (2023), we determine the translation of the body as $transl = [t_x, t_y, f/s]$. The focal length in normalized device coordinate (NDC) space, denoted as $f$, can be computed using the formula $f = 1/tan(FoV/2)$. Here, $FoV$ represents the Horizontal Field of View angle, which is randomly zoomed in from 65 to 25 by following Black et al. (2023).

**Image condition generation.** To synthesize realistic human images with paired pose annotations, we leverage ControlNet, equipped with the state-of-the-art diffusion model, SDXL, as our image generator. Existing ControlNet variants take a 2D skeleton, depth map, or canny map as condition inputs. These inputs are typically detected from real-world images by pre-trained perception models. However, there exist two main drawbacks to generating image conditions from these pre-trained perception models. On one hand, it's laborious to crawl diverse human pose and shape images from the Internet. On the other, the perception models cannot ensure the generation of fully accurate annotations, thus the different modalities annotations have discrepancies, *e.g.*, the 2D keypoint heatmap from a 2D pose estimator and the depth map from a depth estimator are not aligned. In such cases, if we take the perception results as the multi-condition inputs of ControlNet, the generated images would probably have weird artifacts.

To resolve the problem, we construct the input of ControlNet by taking advantage of the 3D human parametric model, SMPL-X. There exist several large-scale human motion capture databases (Mahmood et al., 2019; Black et al., 2023) with diverse body poses and shapes in SMPL-X format. Thanks to the disentanglement of the pose and shape parameters of the SMPL-X model, we can even recombine the two parameters to generate a human mesh that does not exist in the databases. For example, an overweight man doing an extremely difficult yoga pose. Upon getting the simulated camera parameters aforementioned in Section 3.2 and 3D human mesh from SMPL-X, we can render an existing 3D human mesh into the image plane, as such, getting the corresponding depth map, as well as the 2D keypoint heatmap. Notably, the depth map is proven to be crucial to generate accurate body shapes and the keypoint heatmap is helpful to generate accurate hand gestures. In practice, we set the depth condition scale and keypoint condition scale as 0.8 and 0.5 separately.

**Text prompt generation.** The aforementioned image-based multi-condition maps only provide rough control signals of the foreground information. They are not fine-grained enough to determine the gender of the human, as well as the background scenes of the image. Thus, we incorporate a structured text prompt template to handle this issue. In particular, we designed a simple text template as "A {gender} {action} {environment}". The gender and the action of the person are determined by the SMPL-X annotations. The environment is generated by a large language model, *i.e.*, ChatGPT (OpenAI, 2020). To create photo-realistic humans, we also feed negative text prompts, *e.g.*, "ugly, extra limbs, poorly drawn face, poorly drawn hands, poorly drawn feet", to the model.

Finally, we get all of the input conditions of the ControlNet. We apply a total of 40 inference steps for each sample. The resolution of the generated images are all $1024 \times 1024$. The generated images and the input conditions (2D keypoints, SMPL-X parameters) are regarded as the initial data pairs.

### 3.3 Label Denoising and Refinement

The generated images are not always well-aligned with the input conditions. The most common incorrect case is the generated human and the input conditions form a mirror pair. To resolve this problem, we employ an off-the-shelf 2D human pose estimator, Poseur (Mao et al., 2022), to detect the symmetric joints in the image, *e.g.*, left shoulder and right shoulder. If the average precision (AP) between the detected symmetric joints and the condition keypoint map is lower than a threshold $\sigma$, we need to filter this sample from the final dataset. Besides, we also conduct further refinement steps on the initial 2D keypoint condition and SMPL-X pose parameters as follows.

**2D keypoint refinement.** We get the initial 2D keypoints by projecting the 3D joints of SMPL-X into the image plane with the simulated camera parameters. The intuition of the 2D keypoint refinement is that different pose datasets provide different skeleton formats (Sárándi et al., 2023), even though they sometimes share the same joint names. To tackle the label discrepancies, it is necessary to refine the initial 2D keypoints to the formulation of the target 2D pose dataset. Here, we take the COCO dataset as an example to explain the proposed strategy for refining the initial 2D keypoints from SMPL-X model.

Specifically, we leverage a COCO pre-trained keypoint decoder proposed in Mao et al. (2022) to get more accurate 2D keypoint labels. Concretely, we replace the coarse proposals from fully connected layers with the initial 2D keypoints from SMPL-X, and then several deformable cross-attention (Zhu et al., 2021) operations are performed between the image features and keypoint queries to gradually generate the 2D keypoints in COCO format. Compared to the pure pseudo-labeling process, our refinement strategy has more reliable initial keypoint proposals. Thus, our method has a higher upper bound of the final generated 2D keypoint labels.

**3D head pose optimization.** Another common inaccurate case is that the generated human sometimes has a slightly different head orientation compared to the initial SMPL-X pose parameters. To resolve this problem, we leverage a 2D keypoint detector $\mathcal{M}_K$ to get the 2D facial landmarks. Then, we employ EFT (Joo et al., 2021) to optimize the head pose parameters, with camera parameters and other SMPL-X parameters fixed during the optimization.

## 4 Experiments

### 4.1 Datasets and Evaluation Metrics

**Datasets for 3D HPS.** BEDLAM (Black et al., 2023) is a synthetic dataset rendered by Unreal Engine 5, with 1-10 individuals in 8 3D scenes and 95 HDRI panoramas. It offers around 380K unique frames and a total of 1M individual person crops. AGORA (Patel et al., 2021) is another synthetic dataset with 17K images (14K training, 3K test), each image contains 5-15 people in varied lighting or 3D environments. We use them and Pose++ as training sets and perform detailed experiments for a fair comparison. 3DPW (von Marcard et al., 2018) is a in-the-wild dataset will motion capture annotations. It is a standard benchmark in 3D HPS tasks to evaluate the model performance. RICH (Huang et al., 2022) is a dataset aimed at understanding human-scene interactions. We use RICH here as our evaluation dataset for it has various camera views and human-scene contact poses. SSP-3D (Sengupta et al., 2020) is a benchmark dataset designed for body shape prediction methods. It contains 311 images of athletes in form-fitting clothing, showcasing a range of body shapes and poses. We use SSP-3D to evaluate the performance of body shape estimation.

**Datasets for 2D HPE.** COCO (Lin et al., 2014) is a large-scale in-the-wild 2D human pose dataset. We compare the performance of several 2D pose estimators on the COCO training set and Pose++. OChuman (Zhang et al., 2019) is a 2D pose estimation dataset containing various occlusion scenes. We use the OChuman validation set as an evaluation dataset.

**Evaluation metrics.** For 3D HPS, we evaluate the precision of the reconstructed human mesh using 3D evaluation metrics, namely **MPJPE** (Mean Per Joint Position Error), **PA-MPJPE** (Procrustes Analysis Mean Per Joint Position Error), and **PVE** (Per Vertex Error). These metrics calculate the Euclidean distances in millimeters (mm) between the predicted and the actual 3D points or vertices. **PVE-T-SC** (Sengupta et al., 2020) is used as a body shape evaluation metric. For 2D HPE, we adopt widely used **mAP** (mean Average Precision) and its variants as the evaluation metrics.

| Method | Dataset | Output Type | Backbone | 1% Crops↑ | 5% Crops↑ | 10% Crops↑ | 100% Crops↑ |
|---|---|---|---|---|---|---|---|
| RTMPose | C | Classification | CSPNeXt | 0.0 | 7.2 | 23.6 | 67.9 |
| RTMPose | B+C | Classification | CSPNeXt | 46.4 | 55.9 | 58.7 | 68.4 |
| RTMPose | P+C | Classification | CSPNeXt | 49.1 | 55.7 | 58.0 | 68.1 |
| RTMPose | P+B+C | Classification | CSPNeXt | 61.9 | 63.1 | 64.4 | 71.3 |
| RLEPose | C | Regression | ResNet50 | 0.0 | 3.9 | 19.2 | 53.5 |
| RLEPose | B+C | Regression | ResNet50 | 40.6 | 47.7 | 55.3 | 64.8 |
| RLEPose | P+C | Regression | ResNet50 | 31.5 | 39.0 | 50.3 | 65.1 |
| RLEPose | P+B+C | Regression | ResNet50 | 51.8 | 56.3 | 58.5 | 66.6 |

Table 2: Ablation experiments on 2D human pose estimation. C denotes COCO, B denotes BED-LAM, P denotes Pose++, and Crops % only applies to COCO. All experiments are evaluated on the COCO validation set. AP is used as the evaluation metric.

| Method | Dataset | Pretrain | Crops % | PA-MPJPE↓ | MPJPE↓ | PVE↓ | PVE-T-SC↓ |
|---|---|---|---|---|---|---|---|
| CLIFF | B[†] | COCO | 100 | 77.2 | 98.4 | 117.7 | 17.4 |
| CLIFF | B | COCO | 100 | 50.5 | 76.1 | 90.6 | N/A |
| CLIFF | A | COCO | 100 | 54.0 | 88.0 | 101.8 | N/A |
| CLIFF | P | COCO | 50 | 57.6 | 95.7 | 103.4 | 13.7 |
| CLIFF | P | COCO | 100 | 52.7 | 87.3 | 102.1 | 13.4 |
| CLIFF | B+A | scratch | 100 | 61.7 | 96.5 | 115.0 | N/A |
| CLIFF | B+A | ImageNet | 100 | 51.8 | 82.1 | 96.9 | N/A |
| CLIFF | B+A | COCO | 100 | 47.4 | 73.0 | 86.6 | 13.6 |
| CLIFF | P+A | scratch | 100 | 62.3 | 108.7 | 124.1 | 15.4 |
| CLIFF | P+A | ImageNet | 100 | 52.4 | 94.8 | 106.4 | 13.3 |
| CLIFF | P+A | COCO | 100 | 48.6 | 76.8 | 88.9 | 13.3 |

Table 3: Ablation experiments on 3D pose and shape estimation. P denotes Pose++, B denotes BEDLAM and A denotes AGORA. Crops % only applies to *. PA-MPJPE, MPJPE and PVE are evaluated on 3DPW. PVE-T-SC is evaluated on SSP-3D.

## 4.2 ABLATION STUDY

**2D HPE.** In Table 2, we adopt two types of 2D pose regressors to verify the effectiveness of the proposed data generation pipeline. For a fair comparison, all the models are trained with 10 epochs. Our pipeline can consistently improve the detection performance when mixed with different COCO training subsets (from 1% to 100%). The performance of Pose++ is comparable with BEDLAM in all data crops. When joint training with all three datasets, both 2D pose regressors get the best performance. We conjecture that the generated dataset only has one person per image, which lacks human-scene occlusion and human-human interaction. We also find that classification-based RTM-Pose (Jiang et al., 2023) is less data-hungry than regression-base RLEPose (Li et al., 2021a) in low data regime, *e.g.*, achieving much higher AP on 1% COCO training set.

**3D HPS.** In Table 3, we evaluate the impact of Pose++ on a 3D HPS estimator, CLIFF (Li et al., 2022). Pose++ outperforms AGORA Patel et al. (2021) in terms of both pose and shape estimation. Pose++ achieves similar performance when fairly compared with BEDLAM (Black et al., 2023). We conjecture that there still exists some noisy pose labels in our pipeline, which affects the results on 3D pose metrics (PA-MPJPE and PVE). Besides, CLIFF trained on Pose++ achieves better shape estimation results COCO pre-trained backbone on SSP-3D dataset.

**Qualitative Visualization of Control Conditions.** In this section, we demonstrate the necessity of the multi-condition design for generating well-aligned image/annotation pairs. As shown in Fig. 3, When we only use 2D keypoint as a generation condition, the generated human image can be inconsistent with the body shape of the mesh. When we only use rendered depth as a generation condition, the output image may have a different gesture compared to the original SMPL-X mesh. When we use both conditions, both body shape and gesture are aligned with the original mesh. The qualitative experiment verified the effectiveness of the multi-condition design of our generation pipeline.

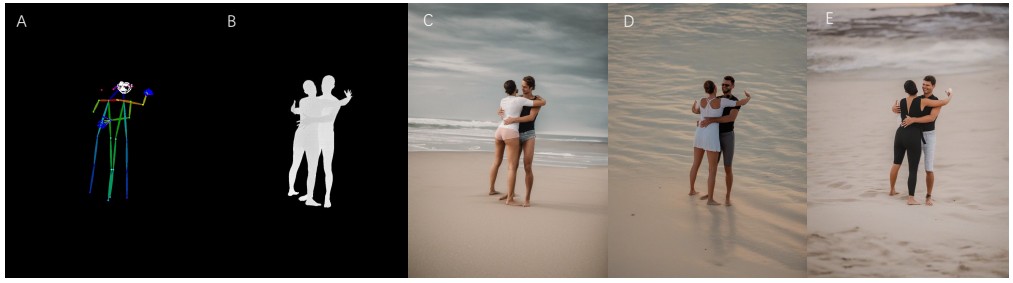

Figure 3: A denotes keypoint condition. B denotes depth condition. C is the generation result with only depth depth condition. D and E are the synthesis results with both keypoint and depth conditions.

## 4.3 MAIN RESULTS

**Results on 2D HPE.** We summaries the key results in Table 4. (1) `Pose++` dataset can improve the result on the COCO validation set. (2) Due to the lack of occlusion and multi-person scenes in the generated images, `Pose++` cannot improve the results on the OCHuman validation set. (3) `Pose++` can outperform DatasetDM Wu et al. (2023) by a large margin on the COCO validation set under the same training setting.

| Method | Backbone | Training Set | Crop | COCO | | | OCHuman | | |
|---|---|---|---|---|---|---|---|---|---|
| | | | | AP | $AP_m$ | $AP_l$ | AP | $AP_m$ | $AP_l$ |
| RTMPose (Jiang et al., 2023) | CSPNeXt | C | 100 | 75.2 | 71.6 | 81.9 | 69.9 | 67.0 | 69.8 |
| RTMPose (Jiang et al., 2023) | CSPNeXt | P + C | 100 | 75.7 | 72.4 | 82.9 | 67.2 | 62.5 | 67.2 |
| SimplePose (Xiao et al., 2018) | HRNet-W32 | C | 100 | 74.9 | 71.3 | 81.5 | 59.8 | 65.3 | 59.8 |
| SimplePose (Xiao et al., 2018) | HRNet-W32 | D + C | 1 | 47.5 | 44.2 | 52.6 | N/A | N/A | N/A |
| SimplePose (Xiao et al., 2018) | HRNet-W32 | P + C | 1 | 50.3 | 44.7 | 59.1 | 29.5 | 18.7 | 29.5 |

Table 4: Main Results on 2D Human Pose Estimation. P denotes `Pose++`, D denotes DatasetDM (Wu et al., 2023), C denotes COCO, B denotes BEDLAM (Black et al., 2023). Crops % only applies to COCO during the training. We evaluate results on COCO and OCHuman Datasets.

**Results on 3D HPS.** Table 5 shows the results on 3DPW and Rich. We report CLIFF Li et al. (2022) trained on `Pose++`, BEDLAM Black et al. (2023) and AGORA Patel et al. (2021). CLIFF trained on `Pose++` shows stronger generation capacity and achieves the best results on both 3DPW and RICH after finetuning on the 3DPW training set.

| Methods | 3DPW (14) | | | RICH (24) | | |
|---|---|---|---|---|---|---|
| | PA-MPJPE↓ | MPJPE↓ | PVE↓ | PA-MPJPE↓ | MPJPE↓ | PVE↓ |
| HMR (Kanazawa et al., 2018) | 76.7 | 130 | N/A | 90.0 | 158.3 | 186.0 |
| SPIN (Kolotouros et al., 2019) | 59.2 | 96.9 | 116.4 | 69.7 | 122.9 | 144.2 |
| SPEC (Kocabas et al., 2021b) | 53.2 | 96.5 | 118.5 | 72.5 | 127.5 | 146.5 |
| PARE (Kocabas et al., 2021a) | 50.9 | 82.0 | 97.9 | 64.9 | 104.0 | 119.7 |
| HybrIK (Li et al., 2021b) | 48.8 | 80 | 94.5 | 56.4 | 96.8 | 110.4 |
| CLIFF[†] (Li et al., 2022) | **46.4** | 73.9 | 87.6 | 55.7 | 90.0 | 102.0 |
| BEDLAM-HMR[*] (Black et al., 2023) | 47.6 | 79.0 | 93.1 | 53.2 | 91.4 | 106.0 |
| BEDLAM-CLIFF[*] (Black et al., 2023) | 46.6 | 72.0 | 85.0 | 51.2 | 84.5 | 96.6 |
| Pose++ CLIFF[*] | 46.6 | **70.2** | **83.7** | **51.0** | **84.4** | **96.1** |
| BEDLAM-CLIFF[*] (with 3DPW) | 43.0 | 66.9 | 78.5 | 50.2 | 84.4 | 95.6 |
| Pose++ CLIFF[*] (with 3DPW) | **42.3** | **65.2** | **76.8** | **50.1** | **82.7** | **93.6** |

Table 5: Reconstruction error on 3DPW and RICH. *Trained with BEDLAM training set. †Trained on real images with same setting as BEDLAM-CLIFF.

**Dataset visualization.** We visualize the generated dataset in Fig. 4. The qualification result demonstrates that `Pose++` can generate diverse human images with well-aligned annotations in the wild. Please refer to Appendix A.1 to see more visualization examples of our dataset.

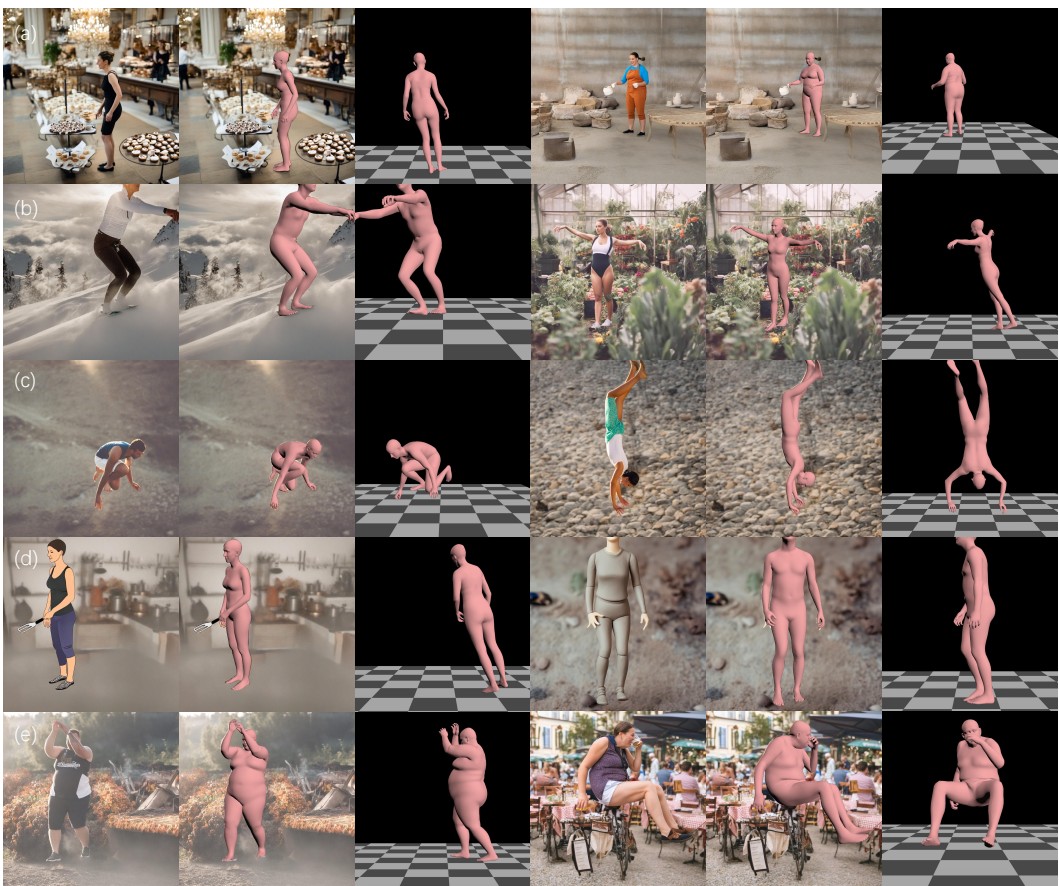

Figure 4: Visual examples of the generated dataset. (a) and (b) demonstrate the diverse scenes of the dataset. (c) indicates the versatile poses of the dataset. (d) illustrates the comic style. (e) shows two examples of overweight body shapes.

## 5    DISCUSS AND CONCLUSION

In this work, we propose an effective data generation pipeline, which can effortlessly generate diverse human images in the wild and corresponding 2D/3D pose annotations with conditional generative models. To further reduce the label noise, we propose to employ an off-the-shelf 2D pose estimator to filter negative samples and optimize the initial pose parameters. We validate the effectiveness of the pipeline on both 3D human mesh reconstruction and 2D human pose estimation. We hope this work could pave the way for using generative models to generate high-quality data for 3D human perception tasks.

**Future work.** Our pipeline can apply to a series of similar tasks where high-quality data pairs are hard to collect, *e.g.*, 3D animal pose estimation and 3D reconstruction of human-object/human-human interaction, by rendering all 3D objects into 2D image conditions, and then generating image-annotation pairs with diffusion models. We leave these promising areas for future work.

**Limitations.** Although our data generation pipeline is cheap and effective, there are a few limitations. First, our data generation pipeline cannot handle crowd scenes, where many humans with small scales are in the image. Second, our pipeline cannot generate video frames since the current design does not consider the consistency of the generated human identities.

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

# A APPENDIX

## A.1 MORE VISUALIZATIONS OF HUMAN INTERACTIONS

We show visualization results on human interactions. Fig. 5 demonstrates that our pipeline can generate well-aligned image-annotation pairs where people are with close interactions. The generated data pairs are of great value in enhancing existing human interaction datasets collected in the studio environment. (*e.g.*, Hi4D Yin et al. (2023) and CHI3D Fieraru et al. (2020))

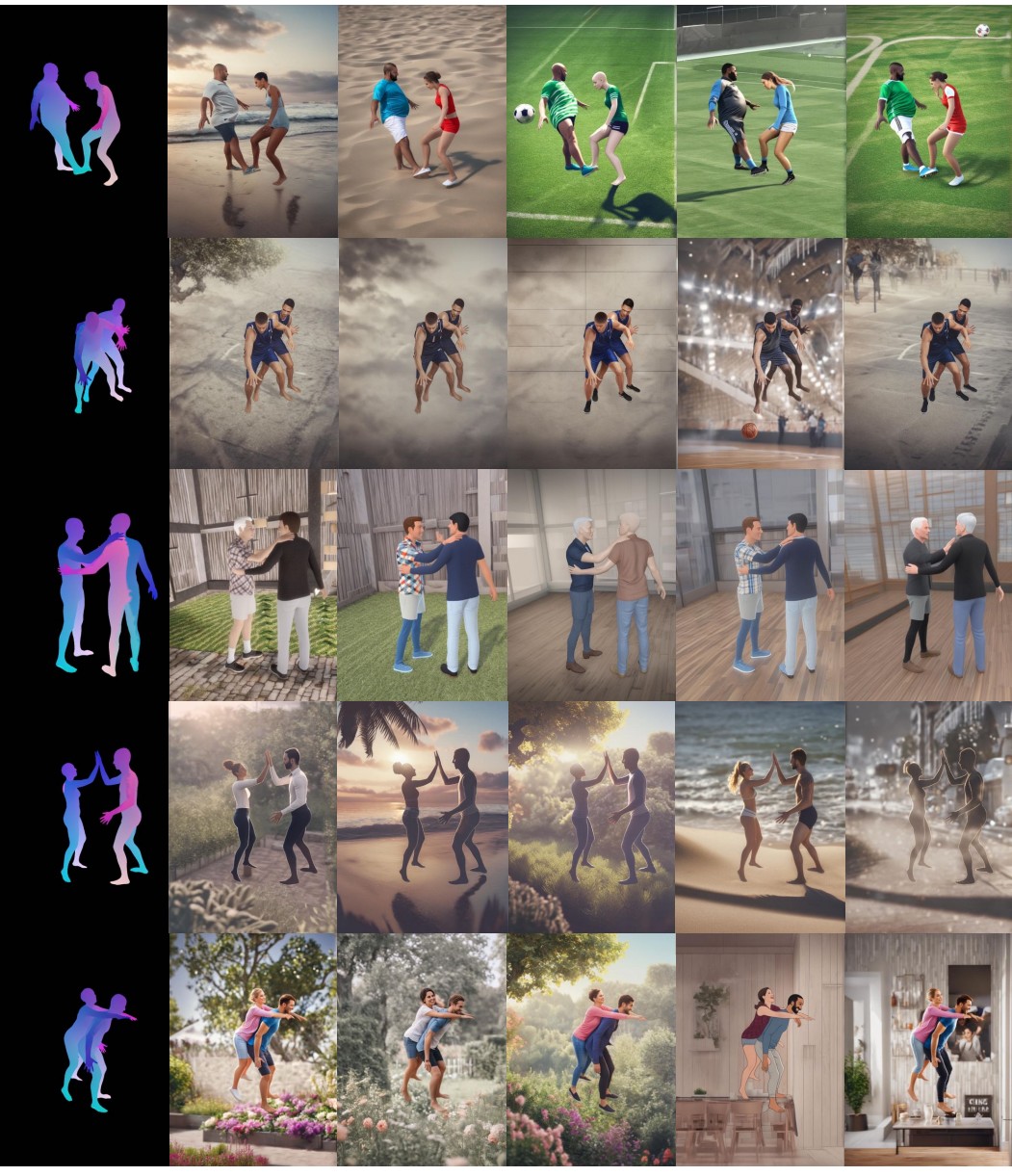

Figure 5: Visualization of Human interaction. The SMPL interaction annotations are sampled from the Hi4D Yin et al. (2023) dataset.

## A.2 FAILURE CASES OF THE INITIAL TRAINING PAIRS

In Fig. 6, we show some failure cases of the initial data pairs generated by ControlNet. The inconsistency of the image and 3D mesh would affect the performance of the 3D human pose estimation. Thus, it is necessary to conduct the label refinement proposed in this work.

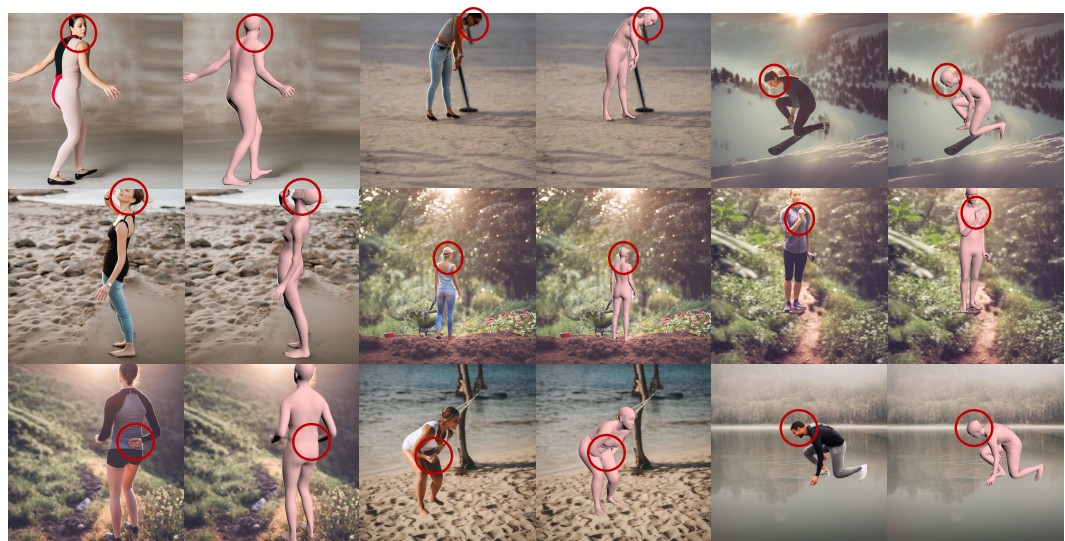

Figure 6: Failure cases of the initial data pairs generated by ControlNet. Highlight with red circles.

## A.3 TEXT PROMPT EXAMPLES GENERATED BY LLM

In Appendix A.3, we show some text prompt examples, which are generated by ChatGPT OpenAI (2023) with diverse human actions and scenes.

| gender | action | environment |
|---|---|---|
| a man | playing soccer | at the park |
| a woman | reading a book | on the beach |
| a woman | dancing | at a nightclub |
| a man | eating dinner | at a restaurant |
| a woman | walking | in the city |
| a woman | swimming | in the pool |
| a man | shopping | at the mall |
| a woman | running | in the park |
| a man | studying | at the library |
| a man | working | at the office |
| a man | chatting | at a cafe |

Table 6: Text prompt examples.

