# OpenReview forum: "3D Human Reconstruction in the Wild with Synthetic Data Using Generative Models"
_ICLR.cc/2024/Conference — ICLR 2024 Conference Withdrawn Submission_

### Official Review · Reviewer_NnQT · 2023-10-14

**Soundness:** 2 fair
**Presentation:** 3 good
**Contribution:** 2 fair
**Rating:** 3
**Confidence:** 5

**Summary:**

This paper proposes a 3D human dataset generation pipeline, motivated by recent advancements of generative AIs. Existing dataset capture pipelines have their own limitations, as introduced in Table 1, which motivates such the proposed automatic dataset generation pipeline. The dataset generation pipeline consists of ControlNet and annotation refinement. The authors showed the benefit of using the proposed dataset in various 3D human reconstruction benchmarks.

**Strengths:**

I mostly agree with the authors’ direction. Using generative AIs, people could generate large-scale 3D human datasets, which are greatly challenging to be captured from real in-the-wild environments.

**Weaknesses:**

1. Image quality is not good. I’m not sure about the training set of ControlNet that takes 2D pose and depth map as inputs, but I suspect the training should be synthetic ones, such as BEDLAM considering the quality of images. Also, paired (2D pose, depth map, image) triplets are greatly hard to get from in-the-wild images, so I think the triplets to train ControlNet should be from synthetic ones (and I think the authors did not train ControlNet, but just use the pre-trained one?). Figure 4 shows that interaction with the background and human is very unnatural and the light consistency lacks.

2. If the experimental results support the effectiveness of the generated dataset, maybe quality of images should not be very important, but unfortunately not. Table 3 shows that the generated dataset is mostly positioned somewhere between AGORA and BEDLAM. Table 5 is confusing. The authors said that the asterisk represents methods trained with the BEDLAM training set. Then what does Pose++CLIFF* (the last row) mean? Is it trained on the BEDLAM training set and Pose++ training set at the same time?

3. Lack of challenging images. To make datasets beneficial, it should cover some challenging cases where existing ones cannot. In this sense, I’m not sure which area the proposed dataset could only cover while previous ones cannot. As the authors said, all images do not have crowded senses, and poses are actually not quite hard. Diverse image appearances are already addressed in AGORA and BEDLAM. In addition, one big drawback of the proposed dataset compared to AGORA and BEDLAM is that the 3D human annotations are just pseudo-GT (not real GT), while AGORA and BEDLAM guarantee that 3D human annotations are geometrically real GT. In this sense, MSCOCO already provides real images with 3D pseudo-GT using NeuralAnnot and EFT.


--- post rebuttal ---
No rebuttal is submitted. I keep my original rating.

**Questions:**

Table 3: what does the dagger mean? Also, there is an asterisk in the caption.

---

### Official Review · Reviewer_iUwf · 2023-10-23

**Soundness:** 4 excellent
**Presentation:** 3 good
**Contribution:** 2 fair
**Rating:** 3
**Confidence:** 4

**Summary:**

This paper introduce a method to augment the dataset to improve the performance of 3D human body fitting. To this end, it proposes to use existing 2D generator model (Stable diffusion) and existing 3D body model and pose dataset. For example, given a background and 3D human body pose (from existing pose data), this paper condition the projection of 3D body model onto the diffusion model, and generate the person image. This paper shows that training with the augmented data can improve the 3D pose fitting performance.

**Strengths:**

This paper is easy to follow since the idea is quite straightforward and practical.
This paper demonstrates a good use-case of existing stable diffusion.

**Weaknesses:**

Augmenting the data by synthesis is already “very” well-known idea to improve the generalizability and accuracy of a prediction model. This paper effectively combines existing 2D generator engine, 3D body model, and pose data in a straightforward way. While this paper put some effort to improve the naturalness of the generation results (e.d., 2D keypoints refinement and 3D headpose optimization), they are all based on the findings from a previous work. Overall, this paper looks not like research work, but more like engineering work to slightly improve the performance without fundamental improvements on a prediction model; and without addressing a core challenge and research problem.

More importantly, this paper does not fully justify why data augmentation for 3D human pose prediction is needed more. For example, why are existing dataset not sufficient? Is the insufficiency coming from the appearance; body poses; or viewpoints? If the insufficiency came from the body pose, why do we need to render more diverse appearance? Without such a deeper analysis, this paper cannot motivate the goal of this paper.

**Questions:**

Stable diffusion has shown quite weakness in generating human appearance (e.g., distorted face and hand). Could you share some rendering failure cases?; and how does such distorted generation affect the 3D prediction results?

**Details Of Ethics Concerns:**

-

---

### Official Review · Reviewer_QSLS · 2023-10-27

**Soundness:** 3 good
**Presentation:** 3 good
**Contribution:** 3 good
**Rating:** 6
**Confidence:** 3

**Summary:**

The paper introduce a pipeline to generate paired training data for 2D HPE and 3D HPS using off-the-shelf ControlNet. Specifically, by projecting the SMPL-X mesh to perspective camera as depth map and 2D keypoints, the method synthesize the image using depth2image and keypoint2image modality in ControlNet. To further compensate the inaccuracy during image generation, 2D pose estimator is leveraged as label filter, and the pose parameters are also fintuned accordingly. Experiments validate that the synthetic data from generative models is helpful for 3D human reconstruction tasks.

**Strengths:**

- The proposed idea to leverage in-the-wild knowledge from generative models is simple but effective.
- The benefit is significant by using synthetic data from the proposed method.

**Weaknesses:**

- The performance boost for 3D HPS in Table 5 seems to be marginal compared with BEDLAM.
- As a data-centric method, I would appreciate more in-depth analysis to the effectiveness of the generated data in helping the existing models. For example:
    - According to Tab. 5, it seems that finetuning with accurate 3D dataset is critical for a good performance. What is the root cause for that?
    - According to Tab. 2, it seems that BEDLAM is consistently better than the proposed data generation pipeline for regression-based backbone (B+C v.s. P+C). However, the proposed data pipeline seems to be more diverse and at scale. What is the root cause for this performance gap?
    - According to the Tab. 3, it seems that the proposed approach achieves a slightly worse results compared with models trained with BEDLAM. The authors’ conjecture is noisy pose labels. I’d like to see a experiment that perturb the labels in BEDLAM to see the results. However, it might be impossible to measure the magnitude of the perturbation to be the same level as in your generated data.
- It is very hard to accurately control the pose of human head and hand using ControlNet. Thus, the majority of noisy labels would come from head and hands in the dataset. I would like to see reconstruction metric specifically for the joint of hands and head.
- Moreover, what if you specifically optimize the pose of hand and head to correct those noisy labels? Global-wise optimization to correct the noisy data might not be enough.

**Questions:**

Please check the weaknesses section.

**Details Of Ethics Concerns:**

This is a paper about human-centric research that leverages the knowledge from the pre-trained diffusion model which is trained on billions of Internet images. It may raise privacy concerns.

---

### Official Review · Reviewer_isbg · 2023-10-29

**Soundness:** 3 good
**Presentation:** 3 good
**Contribution:** 1 poor
**Rating:** 3
**Confidence:** 5

**Summary:**

This paper investigates methods for leveraging the power of diffusion models to generate large-scale paired training data for human pose and shape estimation (HPS). Tips to guide the diffusion model generation process come from two sources. Foreground human pose cues are obtained from existing 3D human motion capture datasets such as AMASS. Text descriptions (including background information) are generated by ChatGPT. ControlNet+SDXL is used as the image generator. They also refined the final annotations through a 2D keypoint-based fitting process. Using synthetic paired training data, some experiments are conducted to verify the effectiveness of their training. The performance has been verified on COCO and OCHuman for 2D HPE, and 3DPW and RICH for 3D HPS. However, no significant performance improvement was observed.

**Strengths:**

1. The resulting image quality looks quite promising. In particular, the ways in which diffusion models and large language models are used are interesting. Post-processing to fit the final annotations to the generated images is reasonable.
2. This exploration would be valuable for the reader to understand how useful current generative models can be for perceptual approaches such as training HPS models.

**Weaknesses:**

1. Experiment. Very limited performance improvements were observed from experiments. This suggests that this synthetic pipeline may not provide sufficiently valuable training samples to bring significant performance improvements to human pose estimation methods. The value of the proposed approach does not seem significant enough.
2. The quality of the generated images is not directly evaluated and compared with other methods. User study may be one of the portable ways to achieve this goal.

**Questions:**

1. The last two lines in the Tab. 4 is very confusing. They used a different crop proportion (1%) compared to the baseline (100%). Therefore, we cannot tell from the table whether synthetic data helps SimplePose using the HRNet-W32 backbone. Additionally, BEDLAM is described in the table title but not in the tab. 4.